# Partial Reduction in *BRCA1* Gene Dose Modulates DNA Replication Stress Level and Thereby Contributes to Sensitivity or Resistance

**DOI:** 10.3390/ijms232113363

**Published:** 2022-11-01

**Authors:** Sandra Classen, Elena Rahlf, Johannes Jungwirth, Nina Albers, Luca Philipp Hebestreit, Alexandra Zielinski, Lena Poole, Marco Groth, Philipp Koch, Thomas Liehr, Stefanie Kankel, Nils Cordes, Cordula Petersen, Kai Rothkamm, Helmut Pospiech, Kerstin Borgmann

**Affiliations:** 1Laboratory of Radiobiology and Experimental Radiooncology, Center of Oncology, University Medical Center Hamburg-Eppendorf, 20246 Hamburg, Germany; 2Project Group Biochemistry, Leibniz Institute on Aging-Fritz Lipmann Institute, 07745 Jena, Germany; 3CF Next-Generation Sequencing, Leibniz Institute on Aging-Fritz Lipmann Institute, Beutenbergstrasse 11, 07745 Jena, Germany; 4CF Life Science Computing, Leibniz Institute on Aging-Fritz Lipmann Institute, Beutenbergstrasse 11, 07745 Jena, Germany; 5Institute of Human Genetics, Jena University Hospital, Friedrich Schiller University, Am Klinikum 1, 07747 Jena, Germany; 6OncoRay-National Center for Radiation Research in Oncology, Faculty of Medicine Carl Gustav Carus, Technische Universität Dresden, Fetscherstr. 74, PF 41, 01307 Dresden, Germany; 7National Center for Tumor Diseases, Partner Site Dresden: German Cancer Research Center, Im Neuenheimer Feld 280, 69120 Heidelberg, Germany; 8Department of Radiotherapy and Radiation Oncology, University Hospital Carl Gustav Carus, Technische Universität Dresden, Fetscherstr. 74, PF 50, 01307 Dresden, Germany; 9Helmholtz-Zentrum Dresden-Rossendorf, Institute of Radiooncology-OncoRay, Bautzner Landstr. 400, 01328 Dresden, Germany; 10German Cancer Consortium (DKTK), Partner Site Dresden, and German Cancer Research Center (DKFZ), Im Neuenheimer Feld 280, 69192 Heidelberg, Germany; 11Department of Radiotherapy and Radiooncology, University Medical Center Hamburg-Eppendorf, 20246 Hamburg, Germany; 12Faculty of Biochemistry and Molecular Medicine, University of Oulu, 90014 Oulu, Finland

**Keywords:** BRCA1, homologous recombination, DNA damage response, CHK1, replication stress, irradiation

## Abstract

*BRCA1* is a well-known breast cancer risk gene, involved in DNA damage repair via homologous recombination (HR) and replication fork protection. Therapy resistance was linked to loss and amplification of the *BRCA1* gene causing inferior survival of breast cancer patients. Most studies have focused on the analysis of complete loss or mutations in functional domains of *BRCA1*. How mutations in non-functional domains contribute to resistance mechanisms remains elusive and was the focus of this study. Therefore, clones of the breast cancer cell line MCF7 with indels in *BRCA1* exon 9 and 14 were generated using CRISPR/Cas9. Clones with successful introduced *BRCA1* mutations were evaluated regarding their capacity to perform HR, how they handle DNA replication stress (RS), and the consequences on the sensitivity to MMC, PARP1 inhibition, and ionizing radiation. Unexpectedly, *BRCA1* mutations resulted in both increased sensitivity and resistance to exogenous DNA damage, despite a reduction of HR capacity in all clones. Resistance was associated with improved DNA double-strand break repair and reduction in replication stress (RS). Lower RS was accompanied by increased activation and interaction of proteins essential for the S phase-specific DNA damage response consisting of HR proteins, FANCD2, and CHK1.

## 1. Introduction

*BRCA1* and *BRCA2* are the most mutated genes in breast cancer [1,2]. For *BRCA1* it was shown that not only a loss but also an amplification of the gene, leading to increased protein expression, result in an adverse prognosis for patient survival due to therapy resistance [3]. 

BRCA1 is one of the critical factors in the DNA repair pathway homologous recombination (HR). It is involved in double-strand break (DSB) recognition via the interaction with abraxas 1, BRCA1 A Complex Subunit (ABRAXAS1), and receptor-associated protein 80 (RAP80) [4]. Subsequently, interaction of BRCA1 with C-terminal-binding protein-interacting protein (CtIP) and the MRE11–RAD50–NBS1 (MRN) complex creates 3′ overhangs of the DNA ends [5]. Replication protein A (RPA) then covers the single-stranded DNA (ssDNA) regions to protect them against nucleolytic degradation until it is replaced by RAD51 to allow strand invasion into the sister chromatid. RAD51 is loaded on to the DNA via the interaction of BRCA1 with PALB2 and BRCA2 [6]. Due to the involvement of BRCA1 in all the key steps of HR, a loss of BRCA1 can lead to a HR-deficiency (HRD) [7,8]. Several studies showed that overexpression of other involved DNA damage response (DDR) or HR proteins can at least partially compensate for the loss and restore the function of HR [9,10]. A defect in HR is associated with an increased sensitivity to platinum-based chemotherapies and PARP1 inhibitors in a synthetic lethal manner, though not always resulting in improved patient survival [5,8,10]. Various HRD scores were proposed to identify an HR defect, most of which employ three independent genetic markers for genomic instability [11]. Some research focuses on the formation of RAD51 foci as a critical indicator [12,13]. A further important function of BRCA1 is the stabilization of stalled DNA replication forks [14,15]. The transient stalling or slowing of replication forks is defined as a main feature of RS [16] and was previously observed in BRCA1 mutant cells [17]. An increased level of RS can cause stalled replication forks to collapse, generating DSBs and subsequently causing genomic instability [18]. RS can be counteracted by elevated CHK1 activation, which inhibits the firing of new origins of replication, allowing stalled replication forks to restart [19,20,21,22]. This may lead to therapy resistance, which was shown to be able to be overcome by CHK1 inhibition in TNBC [9]. A second pathway to overcome RS is via the Fanconi anemia protein complex, mediated by FANCD2/FANCI. Not only in response to DNA damage, but also to increased RS in the cell, FANCD2 and FANCI are ubiquitinated and start to localize at DNA damage foci regions in the nucleus [23]. Together with BRCA1/BRCA2 and RAD51, open DNA ends are protected from degradation by MRE11 or DNA2 [14,24]. Additionally, FANCD2 can protect stalled replication forks independently of FANCI by regulating the complex function of BLM [25]. 

The effects of complete or partial *BRCA1* loss and the importance of defined *BRCA1* mutations in regulatory regions of the gene were investigated. *BRCA1* gene gain or increased protein expression of BRCA1 was considered primarily in the context of resistance to therapy. Many mechanisms are not yet fully elucidated. Mutations located outside functional domains of the BRCA1 protein have not yet been characterized. 

This study focused on understanding how mutations in *BRCA1* exon 9 and 14 influence the resistance against DNA-damaging sources by using a cell line that carries a gain of the BRCA1 gene *BRCA1* gene. *BRCA1* was modified by CRISPR/Cas9-mediated genome editing and the consequent effects on DNA repair processes were assessed by plasmid reconstruction assays and visualized by DNA repair markers such as RAD51, yH2AX, and RPA by immunofluorescence. The expression and activation of DNA damage-response proteins was measured by Western blot. The effects on replication processes were analyzed by DNA fiber assay, cell cycle distribution by FACS analysis, and cellular survival after MMC, talazoparib, and irradiation by colony formation assay.

## 2. Results

### 2.1. BRCA1 Mutations in Exon 9 and 14 Influence DDR Protein Expression and 3D Growth 

Approximately 20% of breast cancer tumors show an amplification of the *BRCA1* gene [1]. Increased expression of BRCA1 was associated with inferior patient survival after therapy [3]. The cause of therapy resistance may be due to the requirement of BRCA1 for efficient DSB repair or superior protection of the DNA replication forks [14]. It is unclear whether mutations in the *BRCA1* gene in an amplified setting influence the therapy resistance. 

Clones of the MCF7 cell line, in which three copies of the *BRCA1* gene are present [26] (Figure 1A and Appendix A), were generated by targeting exon 9 and 14 via CRISPR-Cas9. Clones were carrying indels at the target site detected by analysis of PCR products flanking the target sites (Figure 1B). *BRCA1* exon-9.4 clone and exon-14.2 clone showed large visible indels. All other clones differed from the MCF7 wild type (WT) in a different width of the band. Sequencing and fluorescence in situ hybridization (FISH) analysis of the eight *BRCA1* clones showed complete loss of one allele for all clones. Five clones had at least one mutation in the remaining alleles of *BRCA1* in exon 9 and three in exon 14 (Figure 1A and Appendix A and Table 1). In each of one *BRCA1* exon 9 and 14 clone, all three copies were affected by genetic alteration. Four of the five mutant clones in *BRCA1* exon 9 had the same seven base pair (bp) deletion (c.620del7; p.207fs23stop) leading to a premature stop codon. This was most likely due to microhomology of the target sequence. *BRCA1* exon-9.4 clone showed mutations in both alleles, a 182 bp deletion (c.627del182) leading to a stop at position p.209fs7stop, and an in-frame four amino acid exchange (p.QITP205LLQI). 

The *BRCA1* exon 14 clones showed four different *BRCA1* mutations: *BRCA1* exon-14.1 showed mutations in both alleles (allele 1: 6 bp exchange, leading to a change of two amino acids, p.RW1506GI; allele 2: 14 bp deletion, c.4516 del14; p.1506fs9stop), *BRCA1* exon-14.2 a 183 bp insertion (*c.4517ins183*) that led to a stop at p.1506ins19stop, and *BRCA1* exon-14.3 showed a 7 bp deletion at position c.4511del*7* that also led to an early stop at p.1505fs40stop. The mutations generated most frequently resulted in truncation of the BRCA1 protein in addition to the complete loss of one allele. The effects of the mutations on BRCA1 protein expression were examined by Western blot analysis (Figure 1C). A significant decrease in BRCA1 protein expression was seen in two clones (*BRCA1* exon-9.3 and −9.5), one clone showed a slight increase (*BRCA1* exon-9.4), and all others showed expression levels comparable to WT (Figure 1C). The lymphoblast cell line HCC1937 isolated from a *BRCA1* germline mutation carrier showing residual BRCA1 expression was used for comparison [27,28].

One clone with an affected allele of exon 9 (*BRCA1* exon-9.2) or exon 14 (*BRCA1* exon-14.3) and one with mutations in both remaining alleles (*BRCA1* exon-14.1) were selected. The different clones were analyzed for the expression of other factors involved in the DNA damage response (DDR) (Figure 1D). There was no change in RAD51 expression and a slight decrease in CHK1 in all *BRCA1* clones, but only reaching marginal significance in *BRCA1* exon-9.2 (*p* = 0.051). ATR expression appears to be associated with changes in *BRCA1* exon 14, as both clones showed a reduction compared with the MCF7 WT, with significance in the *BRCA1* exon-14.3 clone (*p* = 0.015). The most striking change was seen in FANCD2, which was significantly reduced by 62–71% in all *BRCA1* clones compared with the MCF7 WT (*p* > 0.0001) (Figure 1D). The growth behavior of all three clones was compared to MCF7 WT, showing clearly visible differences under 3D conditions (Figure 1E). *BRCA1* exon-9.2 showed large, densely clustered, and sharply demarcated spheroids, *BRCA1* exon-14.1 medium, also densely clustered, and *BRCA1* exon-14.3 comparable in size to the MCF7 WT, but with slightly sharper demarcation.

### 2.2. Significant Reduction of HR Capacity Does Not Result in Increased MMC and PARP1i Sensitivity

The effects of the introduced *BRCA1* mutations on the molecular processes that BRCA1 is involved in were examined. BRCA1 is an important factor of HR; the capacity for DSB repair via HR was analyzed. Linearized DNA repair construct plasmid (pDR-GFP) was transfected into cells and repair capacity was detected by flow cytometry (Figure 2A,B). As expected, all *BRCA1* mutated clones showed significantly reduced HR capacity, with a relative reduction of 60% in the *BRCA1* exon-9.2 and -14.1 clones. A reduction of up to 75% in the *BRCA1* exon-14.3 clone was measured compared with the MCF7 WT. Because HR is a cell cycle-dependent process, alterations of the cell cycle profiles of the clones were analyzed (Figure 2C,D). Only minor differences were observed, with an slight increase in G1 and G2/M phase cells in the two *BRCA1* exon 14 clones while the exon 9 clone showed a cell cycle distribution comparable to the MCF7 WT (Figure 2D).

Surprisingly, the sensitivity to mitomycin C (MMC) treatment and PARP1 inhibition (PARP1i) by talazoparib was not enhanced in all *BRCA1*-mutant clones, despite the reduction in HR capacity (Figure 2E,F and Appendix A). Only the two *BRCA1* exon-14 clones showed significantly increased sensitivity to MMC treatment (IC_50_: 0.21 µg/mL and 0.19 µg/mL) compared with MCF7 WT (IC_50_: 0.38 µg/mL) (Figure 2E). However, the *BRCA1* exon-9.2 mutant clone showed a significant increase in MMC resistance (IC_50_: 0.46 µg/mL). The MMC-sensitive *BRCA1* clones with exon 14 mutations were also sensitive to PARP1i with an IC_50_ of 1.3 nM (14.1) and 0.69 nM (14.3), whereas the *BRCA1* exon-9.2 clone was resistant (IC_50_: 6.8 nM) compared with MCF7 WT with an IC_50_ of 4.2 nM (Figure 2F).

### 2.3. Efficient DSB Repair and Rapid DNA Replication Fork Restart Contributes to Therapy Resistance

RAD51 foci formation is a well-established marker to test functionality of HR [12]. Mutations in *BRCA1* are known to impair HR and thereby decrease RAD51 foci formation [29,30]. Therefore, the formation of RAD51 foci in the *BRCA1* mutant clones was examined and quantified (Figure 3A,B). All *BRCA1* clones showed formation of RAD51 foci 6 h after MMC treatment. However, the two (exon 14 clones) MMC-sensitive *BRCA1* showed an approximately 50% lower increase in RAD51 foci compared with MCF7 WT, with respective mean values of 11.5 ± 1.5 (*p* = 0.0093) and 10.1 ± 1.1 (*p* = 0.0026) compared with 18.9 ± 1.7 RAD51 foci per cell. Conversely, the resistant *BRCA1* exon-9.2 clone showed the strongest increase in RAD51 foci with a mean of 22.0 ± 1.1 foci per cell (ns. compared with MCF7 WT). Comparison of RAD51 foci 24 h after MMC showed that both *BRCA1* exon 14 clones also had a reduced ability to resolve the RAD51 foci formed during the observation period, with 10.1 ± 0.9 and 8.4 ± 0.6, respectively. In contrast, resolution of RAD51 foci was most efficient in the *BRCA1* exon-9.2 clone 24 h after MMC treatment, with a significant decrease in RAD51 foci, approximately to the level of MCF7 WT. 

Initiation of HR requires a homology search and pairing of ssDNA, generated by DNA end resection, with the homologous DNA region of the intact strand. Protection of the resected ssDNA is achieved by accumulation of RPA which, as a multifunctional protein, regulates not only ssDNA but also the activity of repair factors [31]. The effect of *BRCA1* mutations in different exons on RPA foci formation 4 h after MMC treatment was determined (Figure 3C and Appendix A). All cell lines showed functionality in the formation of RPA foci. The resistant *BRCA1* exon-9.2 clone showed the highest number of RPA foci per cell with 20.4 ± 1.6 at 4 h, whereas the *BRCA1* exon-14.1 clone after MMC treatment showed the lowest number of RPA foci per cell at 12.3 ± 1.8. In contrast, the *BRCA1* exon-14.3 clone showed a value of 13.1 ± 1.4, comparable to the MCF7 WT, with 14.4 ± 1.7, 24 h after treatment. The strongest decrease in RPA foci was observed in the *BRCA1* exon-9.2 clone, while only a slight decrease occurred in the *BRCA1* exon-14.3 clone; the *BRCA1* clone-14.1 showed almost no change in RPA foci and the MCF7 WT showed an increase to 17.5 ± 1.1 RPA foci per cell. Thus, the behavior of RAD51 foci formation in the cell lines studied was confirmed. Whether the significant differences in the resolution of RAD51 and RPA foci of DNA crosslinks induced by MMC were also reflected by differences in the formation of DSBs was examined. The number of γH2AX foci 6 and 24 h after treatment with MMC was detected and evaluated (Figure 3D and Appendix A). Clear differences between the examined cell lines were observed. The strongest increase in the number of DSB could be observed 6 h after MMC in the resistant *BRCA1* exon-9.2 clone, with 21.1 ± 1.7 compared to the MCF7 WT with 13.6 ± 1.4. The two sensitive *BRCA1* exon-14 clones showed an increase in DSBs compared to the MCF7 WT, with 11.6 ± 0.8 and 15.7 ± 1.3 γH2AX foci per cell after 6 h MMC, respectively. The resistant *BRCA1* exon-9.2 clone showed the most pronounced reduction of DSBs 24 h after treatment, with about 50% compared with only 10% in the MCF7 WT, consistent with the survival (Figure 2E) at corresponding MMC concentration. The *BRCA1* exon-14.3 clone also showed a slight reduction in DSBs, whereas the *BRCA1* exon-14.1 showed no repair of DSBs 24 h after treatment. 

This suggests that in the resistant *BRCA1* exon-9.2 clone, induction of DNA cross-links led to increased DSBs, which, nevertheless, can be repaired most efficiently. Since the resolution of DSBs 24 h after MMC treatment reached a roughly comparable level in all cell lines, differences in DSB repair alone do not seem to reflect sensitivity to MMC. This could be due to differences in BRCA1-dependent stabilization of active DNA replication forks, which prevents nucleolytic degradation of newly synthesized DNA [14,24]. To test this hypothesis, the stability of replication forks after treatment with Hydroxy urea (HU) was examined by a DNA fiber assay (Figure 3E,F). A significant degradation of newly synthesized DNA after HU treatment was observed in MCF7 WT, with 11.6 ± 0.3 μm compared with the untreated control with 12.4 ± 0.4 μm (*p* = 0.045). In contrast, neither the *BRCA1* exon-9.2 nor the two *BRCA1* exon 14 clones showed degradation of previously synthesized DNA. The *BRCA1* exon-9.2 showed no significant effect of HU treatment. The two *BRCA1* exon-14 clones even showed a significant increase in DNA fiber length compared with untreated controls, with 14.9 ± 0.4 vs. 12.3 ± 0.3 μm and 13.9 ± 0.3 vs. 11.7 ± 0.3 μm. It is possible that an increased dNTP level is present in the exon 14 clones. Analysis of replication fork restart after removal of HU represents another feature of functional HR [22]. It was observed that the resistant *BRCA1* exon-9.2 clone had the least difficulty in replication fork restart, visible by the lowest proportion of stalled replication forks, with only 12%, while the two *BRCA1* exon 14 clones showed significantly higher proportions of 15% and 18%, and the MCF7 WT the highest level of about 22% (Figure 3G).

### 2.4. Resistance to Irradiation Emerges from Low Level of DNA Replication Stress

As MMC treatment and PARP1i are mainly causing DNA damage during the S phase; the response to irradiation (IR), inducing various types of DNA damage in all cell cycle phases was analyzed. Cells were irradiated, and cellular survival was determined (Figure 4A). The *BRCA1* exon-9.2 clone was found to be resistant to IR. The *BRCA1* exon-14.3 clone showed significantly increased radiosensitivity, consistent with its sensitivity against MMC treatment and PARPi. The survival of the *BRCA1* exon-14.1 clone after IR was comparable to the MCF7 WT in 2D (Figure 4A). The same distribution of radiosensitivity of the *BRCA1* clones was observed under 3D culture conditions (Figure 4B). However, all cell lines showed lower radiosensitivity in 3D, with significantly smaller differences between cell lines (Figure 4B).

It was tested whether the differences in radiosensitivity affected DNA replication processes. Cells were irradiated with 6 Gy and examined by the DNA fiber assay (Figure 4B,C). Strikingly, the resistant *BRCA1* exon-9.2 clone replicated significantly faster than the MCF7 WT, with a length of 10.1 ± 0.25 µm vs. only 8.7 ± 0.23 µm (Figure 4B and Appendix A). The *BRCA1* exon-14.1 clone also replicated slightly faster than the MCF7 WT with 9.3 ± 0.24 µm. Only the *BRCA1* exon-14.3 clone showed a slowing of replication to 8.4 ± 0.21 µm, already in the untreated situation. Thus, there appears to be endogenously less RS in the resistant *BRCA1* exon-9.2 clone. IR resulted in minimal shortening to 8.1 ± 0.22 µm in the resistant *BRCA1* exon-9.2 clone (Figure 4C). In the clones with mutant exon 14, IdU length was reduced to 6.9 ± 0.24 µm (14.1) and 6.3 ± 0.21 µm (14.3), and the MCF7 WT showed the greatest reduction to 4.9 ± 0.14 µm. This suggests that the resistant *BRCA1* exon-9.2 clone is capable of handling RS induced by IR most effectively. Confirming this, a faster restart of DNA replication, represented by a high value for the IdU/CldU (I/C) ratio, was observed in the resistant *BRCA1* exon-9.2 clone with a value of 0.63 ± 0.02 (Figure 4D). The *BRCA1* exon-14.1 clone showed an I/C ratio of 0.55 ± 0.02, comparable to MCF7 WT with 0.53 ± 0.01, whereas the sensitive *BRCA1* exon-14.3 clone showed the lowest value of 0.48 ± 0.01. 

The cellular mechanism triggering these differences could be the activation of the S phase-specific DDR mediated by ATR and CHK1 (Figure 4E,F). No significant differences were seen in the activation of ATR (Figure 4E), with only MCF7 WT showing an increase in phosphorylation of ATR 6 h after IR, whereas the *BRCA1* clones showed no difference compared to the untreated situation. In contrast, activation of CHK1 showed marked differences (n.s.), with the strongest activation in *BRCA1* exon-9.2 clone, slightly lower activation in the two *BRCA1* exon-14 clones, and the lowest activation in MCF7 WT cells (Figure 4F). FANCD2 expression was examined after IR (Figure 4G) to further analyze a replication conflict due to decreased activation of FANCD2 by monoubiquitination, as described for *BRCA1*-deficient cells [32]. It was apparent that the *BRCA1* exon-14.3 and MCF7 WT showed significantly greater activation after IR compared with the *BRCA1* exon-14.1 and the *BRCA1* exon-9.2 clones. No change in the PARP1 expression was observed after irradiation (Appendix A). Activation of the ATR kinase is mediated by RPA-coated single-stranded DNA (ssDNA) [33]. The effect of inhibition of ATR on RPA foci formation after irradiation was investigated (Figure 4H). The most significant increase in RPA foci in the resistant *BRCA1* exon-9.2 clone (*p* = 0.0066) compared with the other cell lines was observed. Thus, the data presented here indicate that both sensitivity and resistance can be caused by mutations in *BRCA1* and, in sum, are attributable to differences in the management of replication conflicts. 

## 3. Discussion

This study showed that a combination of efficient DNA repair and avoidance of RS results in resistance to different DNA-damaging sources in a *BRCA1* mutated clones of the MCF7 cell line, which carries a *BRCA1* gene amplification, confirming data shown by ref. [26]. The amplification of DDR genes such as *BRCA1* and the possible resulting elevated protein expression, present in about 20% of breast cancer tumors, were associated with therapy resistance [1]. BRCA1 has important functions in HR and DNA replication fork protection [24,34]. Though many studies investigate how *BRCA1* gene amplification or mutations contribute to therapy resistance, many questions remain unanswered. Mutations in *BRCA1* exon 9 and 14, in a *BRCA1* gene-amplified setting, are not yet well characterized, as most investigations focus on the functional domains exclusively. 

All *BRCA1* mutated clones analyzed in our study had impaired HR capacity, but RAD51 foci formation was only reduced in clones mutated in exon 14 (Figure 1 and Figure 2). The interaction of BRCA1 with PALB2 promotes RAD51 loading [35], which seems to be impaired in *BRCA1* exon 14 mutated clones. This might be due to the location of the introduced mutation which is behind the coiled-coil (CC) domain, interacting with PALB2 [36] and the S/T-Q cluster domain (SCD), which harbors the ATR/ATM phosphorylation sites [37]. However, the *BRCA1* exon-9.2 mutated clone can efficiently form RAD51 foci, despite a reduced HR capacity. This mutation is located in between the RING domain, required for BARD1-BRCA1 heterodimerization to perform the E3 ubiquitin ligase activity [38] and the nuclear localization signal (NLS). It was previously shown that mutations in the *BRCA1* gene can generate truncated versions of the BRCA1 protein influencing the resistance to different DNA-damaging agents [39,40,41,42]. It is possible that a truncated protein variant in the *BRCA1* exon 14 mutated clones may contribute to the observed reduced formation of RAD51 foci, as the NLS would still be present on a truncated protein version, but the function of the CC-domain might be affected. This would not be the case for the resistant *BRCA1* exon-9.2 clone. It is well-established that BRCA1 signaling at DSB sites is not restricted to promotion of HR via PALB2–BRCA2–Rad51. The BRCA1-A complex formed on the scaffold protein ABRAXAS1 at the damage site is important for pathway choice for the repair of DSBs [43]. Subsequently, it is the recruitment of CtIP to form the BRCA1-C complex that promotes end resection and HR. Alternatively, recruitment of the helicase BRIP1 (a.k.a. BACH1, FANCJ) is required to form the BRCA1-B complex that is involved in DNA damage response in S phase [44]. The interactions of BRCA1 with ABRAXAS1, BRIP1, and CtIP are all mediated by the BRCA1 carboxyterminal (BRCT) domain binding a phosphorylated by S phase cyclin-dependent kinases. Truncations in exon 14 would thus prevent formation of any of the BRCA1-A, -B, and -C complexes.

HR capacity was determined by a transient transfection of the DR-GFP plasmid since it was shown that a transient or stable transfection does not influence the outcome of the observed HR capacity [45]. It is likely, though, as the introduction of a single or very few DSBs is not sufficient to activate a global DNA damage response [46]. Therefore, analyzing RAD51 foci formation as a readout for the functionality of HR was conducted. The ability of the *BRCA1* exon-9.2 clone to efficiently form RAD51 foci contributes to the observed resistances to PARP1i and MMC treatment as both agents induce DSBs that are repaired by HR [47]. Thus, no complete defect of HR was observed in any of the *BRCA1* mutated clones, though the exon 9.2 mutated clone seemed to be more proficient in HR than the exon 14 mutated clones.

Mechanisms contributing to resistance to PARPi in *BRCA1* mutated cells are (I) increase in drug efflux, (II) restoration of HR, (III) decreased PARP1 trapping, and (IV) stabilization of stalled DNA replication forks [9,48]. All *BRCA1* mutated clones still show residual HR activity, most strongly in the MMC- and PAPR1i-resistant *BRCA1* exon-9.2 clone, contributing to the observed resistance. However, the rather small differences in HR performance strongly suggest that a second mechanism partly responsible for the observed resistance is involved. It is unlikely that a decrease of PARP1 trapping occurred since the resistance was also visible after MMC treatment. Therefore, the most obvious mechanism responsible for resistance seems to be the stabilization of the DNA fork [49], since BRCA1 is also known to play a role in this process. The resistant *BRCA1* exon-9.2 clone indeed showed the most efficient DNA replication fork restart after HU treatment and no degradation of newly synthesized DNA, which suggests that this clone has the most stable replication forks compared with the *BRCA1* exon 14 mutated clones and the MCF7 WT (Figure 3). 

DNA damage induced by MMC treatment and PARP1i occurs predominantly in the S phase of the cell cycle, disrupting the DNA synthesis and leading to DSBs, which are then repaired by HR [50]. Ionizing radiation (IR) was used to test the DNA-damaging source, inducing several DNA damages. IR causes the same results as after MMC treatment and PARP1i. The *BRCA1* exon-9.2 clone was radioresistant, while the *BRCA1* exon-14.3 clone was radiosensitive and the *BRCA1* exon-14.1 clone showed comparable survival to the MCF7 WT (Figure 4). Since the previous results indicated a lower RS in the resistant exon-9.2 clone, the data after IR confirmed these observations. The *BRCA1* exon-9.2 clone showed the highest replication speed, while the 14.3 clone showed the lowest. Since the main feature of replication stress is defined as transient slowing or stalling of DNA replication forks [16], this strongly indicates that the resistant 9.2 clone exhibits the least RS in the untreated state, while the sensitive 14.3 clone exhibits the highest. The same could be observed after IR, as again the *BRCA1* exon-9.2 showed the lowest RS, and the exon-14.3 clone the highest. A high level of RS leads to genomic instability [51] and ultimately increased cell death, which correlates with the survival outcome determined in the colony formation assay. 

The differences in RS were not manifested by differences in the ATR activation after IR, whereas significant differences in the CHK1 phosphorylation were observed, indicating the described ATR-independent activation of CHK1 in the *BRCA1* mutated clones (Figure 4) [52]. CHK1 can counteract RS by inhibiting firing of dormant origins to allow the cell to restart of the stalled forks instead of opening dormant origins [20,21,22,53]. Since the highest activation of CHK1 is observed in the resistant *BRCA1* exon-9.2 clone, this further supports the observation that it has the lowest level of RS. CHK1 phosphorylation is not as high in the *BRCA1* exon 14 mutated clones, not rescuing the already high RS. Supporting this, the strongest accumulation of RPA after inhibition of ATR was seen in the resistant BRCA1-9.2 clone after irradiation (Figure 4).

Overexpression of remaining DDR genes was shown to compensate the loss of some DDR genes such as *BRCA1* [45,54]. However, this seems not to be the case in the present *BRCA1* mutated clones. In the untreated state no elevated expression of RAD51, ATR, or CHK1 could be observed. FANCD2 was even significantly downregulated in all *BRCA1* mutated clones, which is surprising since it has been shown that depletion of *BRCA1* together with *FANCD2* is synthetically lethal [32]. Upon DNA damage induction, FANCD2/FANCI are increasingly ubiquitinated and co-localize with DNA damage foci in the nucleus [55,56], which also happens in response to RS [57]. They protect stalled replication forks together with BRCA1/BRCA2/RAD51 from MRE11 or DNA2 degradation [14,24]. FANCD2 can, however, also act independently of FANCI. It was shown, that FANCD2 can regulate BLM functions to promote the recovery of stalled replication forks [25]. All *BRCA1* mutated clones showed functional FANCD2 ubiquitination after IR, except for the *BRCA1* exon-14.1 clone. This was quite unexpected, as it has a medium level of RS, comparable to the MCF WT, but basically no increase in FANCD2 ubiquitination. The failure of activating FANCD2 in this clone must be further investigated. In the sensitive *BRCA1* exon-14.3 clone, the L/S ratio of FANCD2 (Large/Small FANCD2) was the highest, strongly suggesting that this clone tries to compensate for the high level of RS by high ubiquitination of FANCD2. In the resistant *BRCA1* exon-9.2 clone, a lower L/S ratio was observed, which was expected as this clone has low RS.

In summary, this study showed that *BRCA1* indels in exons 9 and 14 can result in increased therapy resistance or sensitivity. In addition, it seems that a dominant negative effect of the truncating exon 14 mutations is present. All results taken together strongly indicate that a combination of efficient DNA repair and avoidance or rather enhanced counteraction against RS by increased CHK1 activation is responsible for the observed therapy resistance. The analysis of how to overcome this resistance might enable new strategies for treatment of tumors with similar features.

## 4. Materials and Methods

### 4.1. Cell Lines, Culture and Treatment

The MCF7 and HCC1937 cell lines were purchased from the American Type Culture Collection (ATCC, Manassas, VA, USA). All cell lines were cultivated in DMEM medium with 10% FCS and 1% penicillin streptomycin in incubators at 37 °C, 5% CO_2_ atmosphere, and 100% humidity in cell culture flasks. For the 3D growth 10,000 cells were seeded in 25 µL drops of Basement Membrane Extract (Cultrex^®^, growth factor reduced) in 24-well plates, covered with 750 µL culture medium. For mitomycin C (MMC; medac GmbH) treatment concentrations ranging from 1.5 µM to 3.0 µM were used for a maximum of 1 h incubation time. Talazoparib (Selleckchem #S7048) treatment was carried out for 24 h with concentrations of 1 to 10 nM. Ceralasertib (Biozol, Eching, Germany) treatment was carried out for 4 h before and 24 h after irradiation at a concentration of 1 µM.

### 4.2. CRISPR/Cas9-Mediated Modifications of BRCA1

The CRISPR/Cas9 system was used for genome editing of the *BRCA1* gene essentially as described [58]. Two different exons (9 and 14) were targeted with one single-guide RNA (sgRNA) each, designed with the help of crispr.mit.edu (Table 2). An additional guide targeting intron 1 was found to be inefficient and was not followed up further.

The oligonucleotides were hybridized and ligated into vector PX458 (Addgene #48138, kindly provided by I. Vetterlein), digested with *Bbs*I, and amplified in *E.coli* (DH5α). The correct insert with the sgRNAs was confirmed by Sanger sequencing. The sgRNA-expressing plasmids were transiently transfected into MCF7 cells using electroporation. Successfully transfected cells (GFP-positive) were sorted after 48 h via flow cytometry as single cells and expanded in 96-well plates.

### 4.3. PCR Screening

Genomic DNA was isolated using DNeasy Blood & Tissue (Qiagen, Hilden, Germany) and DNA concentrations were measured using a NanoDrop™ One/OneC Microvolume UV-Vis Spectrophotometer (ThermoFisher Scientific, Waltham, MA, USA). Primers were designed to anneal around the Cas9 cutting site within the *BRCA1* gene. To detect small indel variations, a short sequence around the Cas9 cutting site was amplified, while to detect larger variations a longer fragment was amplified (Table 3)

For PCR, GoTaq^®^ Flexi Polymerase (Promega, Walldorf, Germany) with the Green GoTaq^®^ Flexi Reaction Buffer supplemented with 5 mM MgCl_2_, 200 µM dNTPs, 100 pmol/µL forward and reverse primer, and 100 ng genomic DNA as template was used. PCR was performed using a Primus 25 advanced^®^ Thermocycler with the following conditions: Initial denaturation (95 °C for 2 min) followed by 35 cycles of denaturation (95 °C for 30 s), annealing (54 °C for 30 s) and extension (72 °C for 50 s), ending with a final extension (72 °C for 10 min). Fragments were then loaded on to a 2% agarose gel, run for 30 min at 120 V, and stained with ethidium bromide, or analyzed by BioAnalyzer 2100 Expert (B.02.08.SI648) using DNA 7500 chips (Agilent Technologies, Santa Clara, CA, USA). 

### 4.4. Amplicon Sequencing and Clone Characterization

Sequencing of amplicons was performed using Illumina’s next-generation sequencing methodology [59]. Amplicons per clone (Table 3) were pooled, while each pool contained long and short fragments of exon under consideration plus additional fragments for (either) *POLD* (Primers: 5′-CCTGTGCAATTAGGCTTGAG and 5′- CTTCAGGCCGACCTTGAATG; amplicon size 500 bp) or *POLE* (Primers: 5′-GGTGTTCAGGGAGGCCTAAT and 5′-TACTTCCCAGAAGCCACCTG; amplicon size 195 bp) serving as controls. Amplicons were quality checked and quantified using the 2100 Bioanalyzer instrument in combination with a high-sensitivity DNA kit (both Agilent Technologies). Prior to library preparation, amplicons per clone were pooled as described above. Libraries were prepared from 50 ng of pooled amplicons (per clone) using NEBNext Ultra II Directional RNA Library Preparation Kit in combination with NEBNext Multiplex Oligos for Illumina Set 1 (96 Unique Dual Index Primer Pairs) following the manufacturer’s instructions in general (New England Biolabs, Frankfurt am Main, Germany). Deviating from the protocol, the amplicons were not fragmented but inserted directly into the library preparation. Quantification and quality checking of libraries was conducted using a 2100 Bioanalyzer instrument and DNA 7500 kit (Agilent Technologies). Libraries were pooled and sequenced on a MiSeq (Illumina, San Diego, CA, USA). System runs in 301 cycle/paired-end mode using SBS 600 cycles v3 sequencing reagents. Sequence information was converted to FASTQ format using bcl2fastq v2.20.0.422.

Both adapter clipping and quality trimming were applied to the raw reads using Cutadapt 2.8 [60] (parameters: -q 15 -m 1 -a AGATCGGAAGAGCACACGTCTGAACTCCAGTCA -A AGATCGGAAGAGCGTCGTGTAGGGAAAGAGTGT). The resulting paired-end reads were merged with the tool fastq_mergepairs from USEARCH v8.0.1517 [61]. Amplicon sequences were identified in the merged reads based on their primer sequences with the Python script identifyAmplicons.py (parameter: --primers primers.fa) (https://github.com/PhKoch/amplicon/releases/tag/0.3, accessed on 28 September 2022). For each clone, the abundance of the amplicons was determined to characterize its type of mutation. To this end, sequences of amplicons with high abundance were aligned to the genomic *BRCA1* sequence. Only two different amplicons with comparable abundance were identified in all clones analyzed (Table 1 and Appendix A); despite the expected presence of three genomic copies of *BRCA1* in MCF7, the clones were further characterized by FISH.

### 4.5. FISH Analysis

FISH was used according to standard procedures and previously described [62]. Two commercially available probes were combined with a homemade one: a BRCA1-specific probe (Abnova, Heidelberg, Germany; SpectrumOrange), a centromere-specific one for chromosome 17 (D17Z1–Abbott/Vysis, Chicago, IL, USA; SpectrumGreen), and a whole chromosome paint for chromosome 17 (wcp 17; SpectrumAqua) [62]. Then, 10 to 20 metaphases per cell line were acquired and analyzed for each probe set on a Zeiss Axioplan microscope, equipped with ISIS v2.86 software (MetaSystems, Altlussheim, Germany).

### 4.6. Western Blot and Immunostaining

Total protein was extracted from exponentially growing cells and 25 µg protein was resolved by SDS-PAGE using 4%–15% gradient gel (Bio-Rad Laboratories, Feldkirchen, Germany). After transfer and blocking in 5% BSA for at least 1 h, proteins were detected by primary antibodies against ATR [C-1] (Santa Cruz Biotechnology, Heidelberg, Germany 1:750), p-ATR S428 (Cell Signaling Technology, Leiden, The Netherlands, #2853, 1:500), BRCA1 [MS110] (Calbiochem/Merck Chemicals, Darmstadt, Germany, 1:1000), CHK1 [2G1D5] (Cell Signaling, 1:750), p-CHK1 S296 (Cell Signaling #2349, 1:750), FANCD2 [FI17] (Santa Cruz, 1:2000), PARP1 [C210] (BD, 1:1000), HSC70 [B6] (Santa Cruz, 1:10.000), RAD51 [PC130] (Calbiochem, 1:1000), and β-Actin [AC-74] (Sigma-Aldrich, Taufkirchen, Germany, 1:20.000). Primary antibodies were detected with IRDYE 680 conjugated anti-mouse IgG, IRDYE 800 conjugated anti-mouse IgG, IRDYE 680 conjugated anti-rabbit IgG, or IRDYE 800 conjugated anti-rabbit IgG (LiCor, 1:7500).

For immunofluorescence staining, cells were seeded on culture slides. After treatment, cells were permeabilized, fixed with 4% paraformaldehyde, and blocked overnight in 3% BSA. Foci were detected using primary antibodies against RAD51 [AB-1] (Calbiochem, 1:500), γH2AX [Ser139] (Millipore/Merck Chemicals, Darmstadt, Germany, 1:250) or RPA [MA-34] (Santa Cruz, 1:400), followed by secondary antibodies Alexa Fluor 488 goat anti-rabbit IgG (Cell signaling, 1:600), Alexa Fluor 488 goat anti-mouse IgG (Cell signaling, 1:600), or Alexa Flour 594 goat anti-mouse IgG (Cell signaling, 1:500). Nuclei were stained with DAPI and samples were mounted (Vector Laboratories, Newark, CA, USA). The γH2AX foci were quantified manually by capturing fluorescence images using a Zeiss Axioplan 2 fluorescence microscope equipped with a charge-coupled device camera and Axiovision software (Carl Zeiss Microscopy, Oberkochen, Germany), followed by quantification using Image J software. RPA and RAD51 foci were quantified automatically by the Aklides^®^-system (MediPan, Blankenfelde-Mahlow, Germany). A minimum of 100 cells per dose and slide were quantified.

### 4.7. Homologous Recombination Assay

The homologous recombination (HR) capacity was measured by transient transfection of I-Sce-1 linearized pDR-GFP plasmid (Addgene #264752, kindly provided by M. Jasin). First, 1 µg of the linearized plasmid was transfected into cells using FuGENE (Roche, Mannheim, Germany) in a 1:3 µg/µL ratio according to the manufacturer’s instructions. To measure transfection efficiency, cells were transfected with 1 µg pEGFP-N1 (Addgene #6085-1) in a parallel approach. After 48 h, cells were harvested and the fraction of GFP-positive cells was determined by flow cytometry. HR capacity was calculated according to GFP-positive cells (pDR-GFP) and transfection efficiency (pEGFP-N1). 

### 4.8. Cell Cycle

For cell cycle analysis, cells were harvested, fixed with ice-cold 80% ethanol, and stored at −20 °C. Cells were washed in PBS + 0.1% Tween20 and stained with propidium iodide (10 µg/mL with 1% RNase and 0.1% Triton X-100) for 30 min in the dark. Flow cytometry analysis was performed using a MACSQuant10 with MACSQuantify Software 2.11 (Miltenyi Biotec, Bergisch Gladbach, Germany). The proportion of cells in the respective cell cycle phases was calculated using ModFit LT™ 3.2 software (Verity Software House, Topsham, ME, USA).

### 4.9. 2D and 3D Clonogenic Survival

For the 2D colony formation assay, 250 cells per well were seeded in a 6-well plate 12 h before treatment with MMC or irradiation and were cultured for 14 days. Cells were fixed with 70% ethanol and stained with 1% crystal violet (Sigma-Aldrich). Colonies with more than 50 cells were counted and normalized to untreated samples. 

The 3D soft agar colony formation assay was performed as described before [63,64] with slight variations. For the assay, 96-well plates were used, containing 60 µL 0.6% agarose as bottom layer and 50 µL 0.3% agarose cell-containing top layer with 1500 cells per well. Instead of a feeding layer, 10 µL of medium was additionally added on top of the cell-containing layer. Cells were irradiated 12 h after plating and cultured for 10 days. Colonies with a diameter greater than 50 µm were counted and normalized to untreated controls. Each survival curve represents the mean of at least three independent experiments.

### 4.10. DNA Fiber Assay

Classical DNA Fiber Assay: Exponentially growing cells were pulse labeled with 25 µM CldU (Sigma-Aldrich) followed by 250 µM IdU (Sigma-Aldrich) for 30 min each. HU (2 mM) was given for 4 h in between the labels. The cells were harvested and DNA fiber spreads were prepared and stained as described previously [65]. Fibers were examined using Axioplan 2 fluorescence microscope (Carl Zeiss Microscopy). CldU and IdU tracts were measured using ImageJ software. At least 100 fibers per sample and independent experiment were analyzed.

Coming DNA Fiber Assay: Exponentially growing spheroids at day 3 post seeding were pulse labeled with 25 µM CldU (Sigma-Aldrich), followed by 250 µM IdU (Sigma-Aldrich) for 30 min each. Irradiation with 6 Gy was performed between the first and second label. Spheroids were harvested and DNA fiber was prepared using the DNA combing system of Genomic Vision. Fibers were examined using an Axioplan 2 fluorescence microscope (Carl Zeiss Microscopy). CldU and IdU tracts were measured using Image J software. At least 100 fibers per sample and independent experiment were analyzed. 

### 4.11. Statistical Analysis

Statistical analysis, curve fitting, and graph creation were performed using GraphPad Prism (Version 6.02) software (Graph Pad Software, San Diego, CA, USA). Data are given as mean (+SEM) of at least three replicate experiments. Unless stated otherwise, significance was tested by Student’s *t*-test.

## Figures and Tables

**Figure 1 ijms-23-13363-f001:**
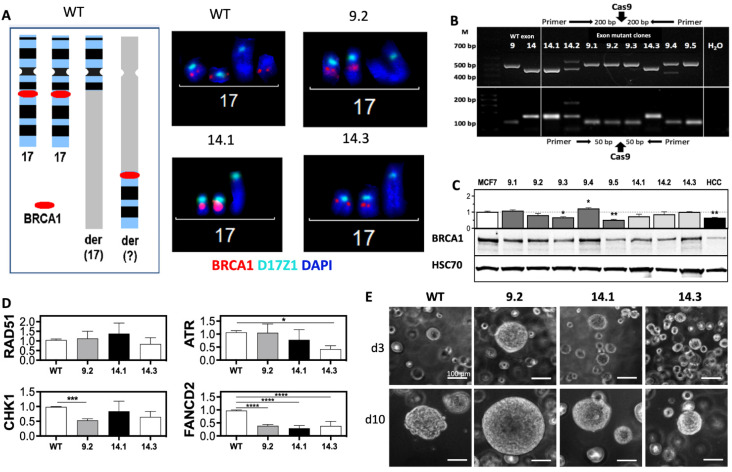
*BRCA1* mutations in exons 9 and 14 alter protein expression and 3D growth. (**A**) FISH analysis of *BRCA1* mutated MCF7 clones shows loss of one of the three BRCA1-alleles. (**B**) PCR screen of CRISPR/Cas9-targeted locus BRCA1 in exon 9 and 14 in MCF7 clones. Fragments with an amplicon size of 494 bp/101 bp (exon 9) and 419 bp/169 bp (exon 14) were selected for detection of large/small indels. PCR products were separated by electrophoresis. (**C**,**D**) BRCA1, RAD51, FANCD2, CHK1, and ATR expression from total cell extracts of exponentially growing cells were separated and analyzed by Western blot and normalized to MCF7 WT. HCC1937 (HCC) served as negative control and HSP70 and ß-ACTIN as loading controls. (**E**) 3D growth of *BRCA1* clones mutated in exon 9 or 14. Single cells were embedded in BME, and spheroids were photographed three (d3) and ten (d10) days after seeding. Scale bars represent 100 µm (20× magnification). Shown are the mean values of three independent experiments ± SEM. Asterisks (*) indicate significant differences (* *p* < 0.05; ** *p* < 0.01; *** *p* < 0.001; **** *p* < 0.0001 Student’s *t*-test).

**Figure 2 ijms-23-13363-f002:**
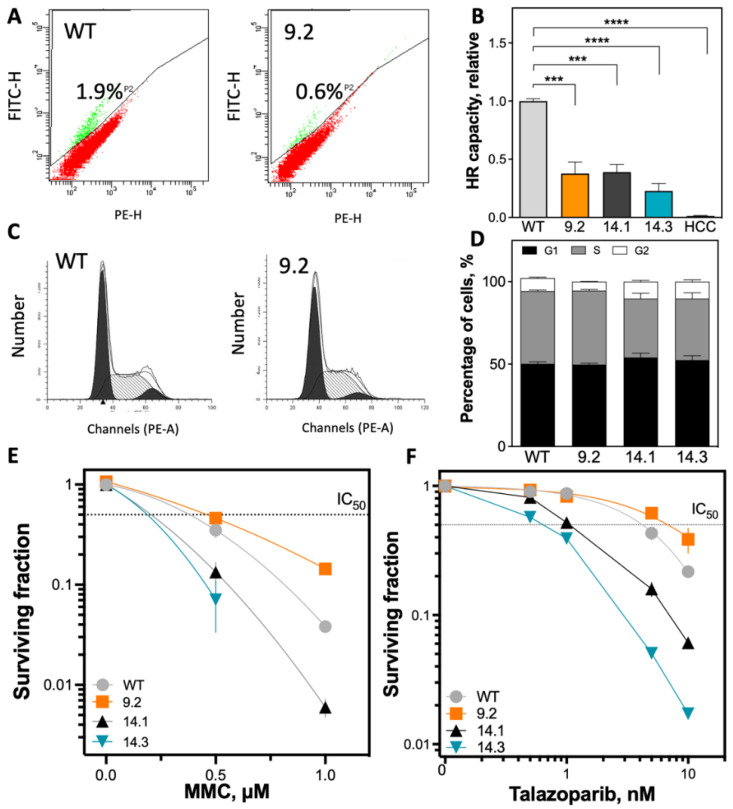
Significant reduction in HR capacity is not reflected by decreased survival after MMC treatment and PARP1 inhibition. (**A**,**B**) *BRCA1* mutant MCF7 clones showed significantly reduced HR capacity in the plasmid reconstruction assay. Cells were transiently transfected with the pDR-GFP plasmid and HR-competent cells (GFP-positive) were detected by flow cytometry. The determined HR capacities were normalized to the MCF7 WT cell line. The *BRCA1*-deficient cell line HCC1937 served as negative control. (**C**,**D**) No differences were detected in cell cycle profiles among *BRCA1*-mutant clones. Cells were stained with propidium iodide and then subjected to flow cytometry analysis. (**E**,**F**) Differences in cellular survival after treatment with MMC or PARP1 inhibition. Cells were seeded 12 h before treatment, treated for 1 h with MMC or 24 h with talazoparib, fixed after 14 days, and the number of colonies with >50 cells was determined. Shown are the mean values of three independent experiments ± SEM. Asterisks (*) indicate significant differences; (*** *p* < 0.001; **** *p* < 0.0001, ns = not significant Student’s *t*-test).

**Figure 3 ijms-23-13363-f003:**
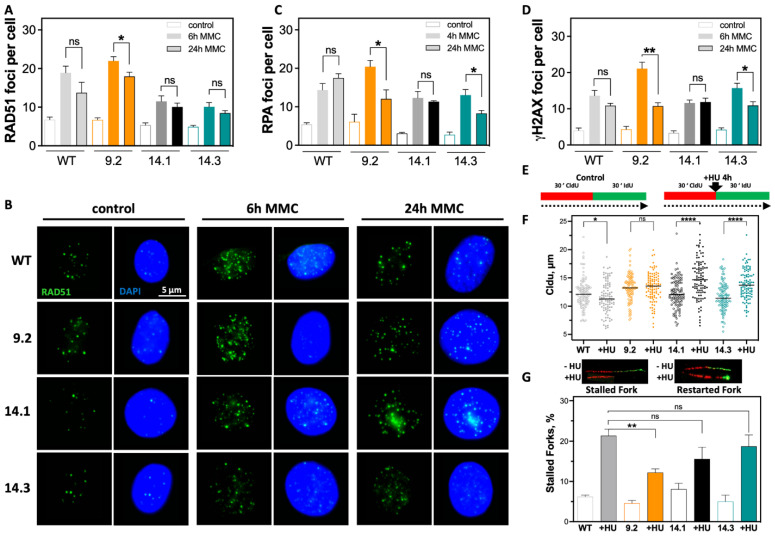
Resistance indicates more efficient DSB repair and avoidance of replication-associated DNA damage. (**A**–**D**) Formation of DNA repair foci after MMC treatment. (**A**,**B**) RAD51 foci (green) 6 h and 24 h after treatment with 0.5 µg/mL MMC for 1 h, (**C**) quantification of RPA foci 4 h and 24 h after MMC, and (**D**) γH2AX foci 6 h and 24 h after using the same protocol. Nuclei were counterstained with DAPI. γH2AX foci were quantified manually; RAD51 and RPA foci were quantified using the Aklides^®^ NUK system (MediPan). Scale bar represents 5 µm (40× magnification). At least 100 cells were analyzed in each biological replicate. (**E**–**G**) DNA fiber assay after treatment with HU for 4 h, in MCF7 WT and *BRCA1* mutated clones. (**E**) Treatment scheme shows sequential labeling with CldU and IdU for 30 min each and addition of HU for 4 h between CldU and IdU. Incorporated nucleotides were detected by immunofluorescence. The length of the DNA fibers was measured with the ImageJ 1.52n software. (**F**) DNA fiber length of DNA strands already synthesized before HU administration and (**G**) number of replication forks halted within the next 30 min after HU removal. Shown are the mean values of three independent experiments ± SEM. Asterisks (*) indicate significant differences (* *p* < 0.05; ** *p* < 0.01, **** *p* < 0.0001; ns = not significant Student’s *t*-test).

**Figure 4 ijms-23-13363-f004:**
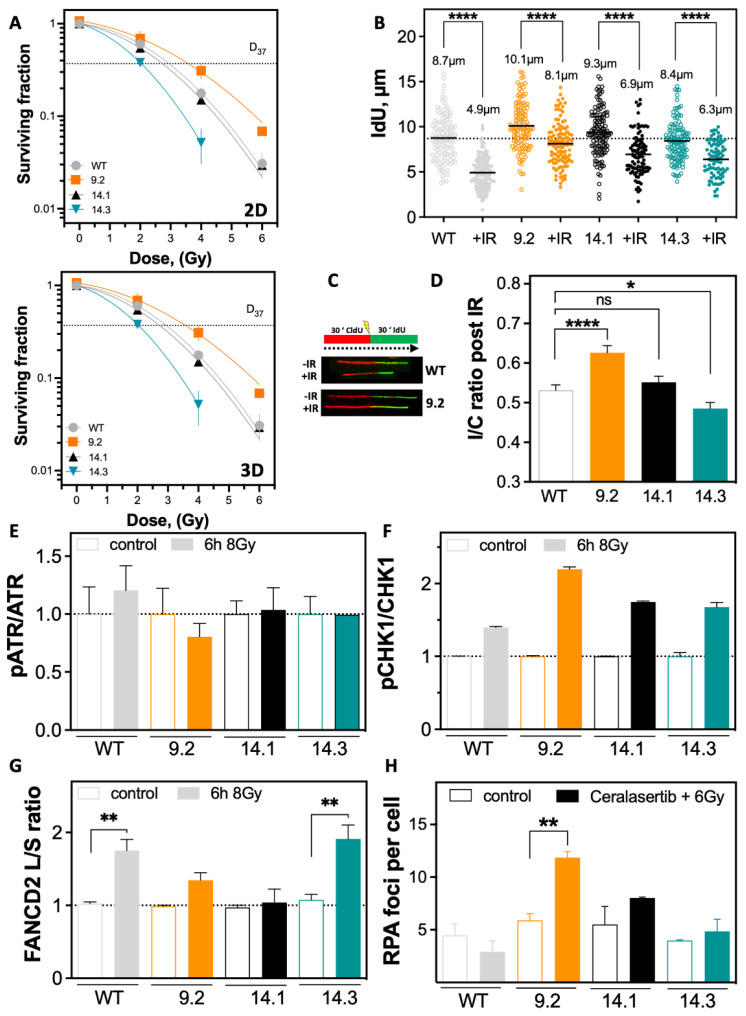
Less DNA replication stress is associated with resistance to irradiation. (**A**,**B**) Differences in cellular survival after irradiation in 2D and 3D cultures. Cells were seeded 12 h before IR, fixed after 14 days, and the number of colonies with >50 cells was determined. (**B**–**D**) DNA fiber assay after irradiation with 6 Gy, in MCF7 WT and *BRCA1* mutated clones. (**B**) Treatment scheme shows sequential labeling with CldU and IdU for 30 min each and IR between both labels. Incorporated nucleotides were detected by immunofluorescence. The length of the DNA fibers was measured with the Image J 1.52n software. (**C**) DNA fiber length of DNA strands after IR (IdU) and (**D**) the ratio of DNA fiber length pre to post irradiation (I/C ratio) was calculated. (**E**–**G**) Activation of ATR, CHK1, and FANCD2 6 h after IR were detected in total cell extracts of exponentially growing cells, analyzed by Western blot and calculated as the ratio of pATR to ATR, pCHK1 to CHK1 or FANCD2L/FAND2S. (**H**) RPA foci after IR and treatment with the ATR inhibitor Ceralasertib were quantified using the Aklides^®^ NUK system (MediPan) at a magnification of 40×. At least 100 cells were analyzed in each biological replicate. Shown are the mean values of three independent experiments ± SEM. Asterisks (*) indicate significant differences; (* *p* < 0.05; ** *p* < 0.01; **** *p* < 0.0001, ns = not significant Student’s *t*-test).

**Table 1 ijms-23-13363-t001:** *BRCA1* Mutations introduced in MCF7 cells via CRISPR/Cas9.

Cell Line	Allele 1	Allele 2	Allele 3
MCF7 WT	WT	WT	WT
MCF7 9.1	WT	7 bp deletion (c.620del7 → p.207fs23stop)	loss
MCF7 9.2	WT	7 bp deletion (c.620del7 → p.207fs23stop)	loss
MCF7 9.3	WT	7 bp deletion (c.620del7 → p.207fs23stop)	loss
MCF7 9.4	In frame 4 aa exchange (p.QITP205LLQI)	182 bp deletion (c.627del182 → p.209fs7stop)	loss
MCF7 9.5	WT	7 bp deletion (c.620del7 → p.207fs23stop)	loss
MCF7 14.1	6 bp exchange (p.RW1506GI)	14 bp deletion (c.4516 del14 → p.1506fs9stop)	loss
MCF7 14.2	WT	183 bp insertion (c.4517ins183 → p.1506ins19stop)	loss
MCF7 14.3	WT	7 bp deletion (c.4511del7 → p.1505fs40stop)	loss

List of mutations in the 3 *BRCA1* wild type (WT) alleles in the selected clones at the coding DNA (c.) and protein (p.) levels with associated position that result in complete loss (loss), deletions (del), and insertions (ins) of base pairs (bp) and cause frame shift (fs) or stop.

**Table 2 ijms-23-13363-t002:** Oligonucleotides used for expression of sgRNA sequences targeting BRCA1. Target sequence is underlined.

Exon	Direction	Sequence (5′ to 3′)
9	forward	CACCGTTGTTACAAATCACCCCTCA
	reverse	AAACTGAGGGGTGATTTGTAACAAC
14	forward	CACCGCCCATCATTAGATGATAGG
	reverse	AAACCCTATCATCTAATGATGGGC

**Table 3 ijms-23-13363-t003:** Primer sequences used to amplify the CRISPR/Cas9 targeted regions of the *BRCA1* gene.

Exon	Length and Direction	Tm (°C)	Sequence (5′ to 3′)	Amplicon Size (bp)
9	short forward	58.9	TTCCCTATAGTGTGGGAGATCA	101
	short reverse	54.7	CAAACTTTGCCATTACCCTTTT	
	long forward	55.3	CCACACCCAGCTACTGACCT	494
	long reverse	55.3	CTCTTCCAGCTGTTGCTCCT	
14	short forward	55.9	CGATGGTTTTCTCCTTCCATT	169
	short reverse	55.3	TTGCTCCTCCACATCAACAA	
	long forward	61.4	CCACACCCAGCTACTGACCT	419
	long reverse	59.4	CTCTTCCAGCTGTTGCTCCT	

## Data Availability

The Amplicon sequencing data discussed in this publication have been deposited in NCBI’s Sequence Read Archive (PMID: 34850941) and are accessible through BioProject accession number PRJNA885589.

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
