# Peer review of "Partial Reduction in BRCA1 Gene Dose Modulates DNA Replication Stress Level and Thereby Contributes to Sensitivity or Resistance"

_ijms, 2022, doi:10.3390/ijms232113363_

Round 1

Author Response

Major comments:

Reviewer comment: The authors had to consider synchronizing the cells and assessing apoptosis or other replication stress markers, such as RPA-2 and/or 53BP1. Evaluation of ubiquitination and PARP-1 expression, determination of deoxyribonucleotide triphosphates (dNTPs) content in the clones, and the employment of ATR-selective inhibitors are also suggested.

Answer: We thank the reviewer for the very helpful suggestions, which were also open for discussion for us.

We considered to perform cell synchronization experiments, but finally restrained from doing so for the following reasons: 1) Most synchronization procedures such as double thymidine block occur at the G1/S boundary or beginning of S phase and themselves already lead to DNA damage and cellular stress. 2) We performed synchronization trials with MCF7 cells using Cdk1 and or Cdk4/6 inhibitors that should not cause DNA damage but found poor synchrony despite attempts to optimize the conditions, probably due to the rather long doubling times of MCF7.  For this reason, we have rather chosen agents that damage mainly in S phase, such as MMC, which induces cross-links, and therefore lead to replication conflicts. In this respect, we assessed the differences in cellular survival after MMC as the first indication of differences in replication stress (as on page 10, 4th paragraph). This observation is also indirectly supported by the same spectrum of sensitivity observed after PARP inhibition (Figure 2F), even though PARP also plays an active role in alternative NHEJ.

We had refrained from analyzing radiation-induced apoptosis in our cell system because MCF7 cells exhibit p53-independent resistance to IR-induced apoptosis and are characterized by the loss of caspase-3 (Essmann et al., Cancer Research 2004).

As suggested by the reviewer, we analyzed the effects of the introduced BRCA1 mutations on PARP expression and observed no differences in expression (see Supplementary Fig. XX and text).

We agree that changes in the dNTP pool would be a possible explanation for some of the observations. However, we also see other possible causes, such as differences in the use of DNA damage tolerance pathways or DNA damage signaling. We sought to investigate the possible effects of differences in nucleotide provision on cellular sensitivity by depleting the nucleotide pool by treatment with hydroxyurea. In doing so, we observed that the resistant clone recovered faster compared to the other cell lines (Figure 3G).

In order to further substantiate our observations, we performed inhibition of ATR in our cell system as suggested by the reviewer and analyzed the number of RPA foci as a marker of increased DNA damage 24 hours after irradiation. We have expanded Figure 4 with the corresponding new data set and added it in the text (Page 8, last Para).

Reviewer comment: In addition, the recruitment of BRCA1 in the DNA damage response may be dependent or independent to PALB2-BRAC2-Rad51. This should be at least discussed and/or mentioned in the text.

Answer: We completely agree with the reviewer in this aspect and have added a paragraph in the discussion section (p. 9, lines 317-326) to address this issue. We are aware that also other functions of BRCA1, such as regulation of transcription may contribute as well. As these were not subject of the present study, we restrained from discussing them here.

Reviewer 1 minor comments:

-Fig.1 A: It is too small; I would suggest to increase the size.

-Fig. 3B: Please, provide the scale bar.

-Fig. 3G: It is too small; I would suggest to increase the size.

Answer: We agree with the reviewer and tried to improved Figure 1, 2 and 3 for better readability and added the missing information

Reviewer comment: -There is a discrepancy between pag. 6, lines 199-200 and pag. 13, lines 528-530. The quantification is not clear. Please, modify it.

Answer: We apologize and remove the discrepancy

Reviewer comment: -Pag. 6, line 210: Please, provide a reference.

Answer: a reference is now provided.

Reviewer comment:

-Pag. 9, line 301: What is Figure X?

Answer: We apologize for this mistake and exchange it to Figure supplemental Figure 5C

-Pag. 9, lines 314-321: I would suggest to remove this paragraph because it is unclear how the authors can make this claim when there is NS difference between the groups.

We have added data in the revised version of the manuscript to support this part of the discussion (Figure 4H) and now added the following sentence in the discussion:

Supporting this, the strongest accumulation of RPA after inhibition of ATR was seen in the resistant BRCA1-9.2 clone after irradiation (Figure 4).

Reviewer comment: Pag. 9, lines 330-332: The sentence starts as “ This study shows…” but at the end the authors provided a reference. So, what is new here?

Answer: We apologize for this misleading sentence which is now changed to:

“This study shows, that a combination of efficient DNA repair and avoidance of RS results in resistance to different DNA damaging sources, in BRCA1 mutated clones of the MCF7 cell line, that carries a BRCA1 gene amplification, confirming data shown by [26].

Reviewer comment: I would suggest to supplement the Tables with a brief relative legend.

Answer: We thank the reviewer for this helpful suggestion and now introduced a brief legend below table 1: List of mutations in the 3 BRCA1 wild type (WT) alleles in the selected clones at the DNA (c.) and protein (p.) levels with associated position that result in complete loss (loss), deletions (del), and insertions (ins) of base pairs (bp) and cause frame shift (fs) or stop.

Reviewer 2 Report

Review Report for the Manuscript “Partial reduction in BRCA1 gene dose modulates DNA replication stress level and thereby contributes to sensitivity or resistance”

Rating the Manuscript

Originality/Novelty: Is the question original and well defined? Do the results provide an advance in current knowledge?

Yes, using a cell line that carries a gain of the BRCA1 gene BRCA1 gene, the authors have studied how mutations in BRCA1 exon 9 and 14 influence the resistance against DNA damaging sources. 

Significance: Are the results interpreted appropriately? Are they significant? Are all conclusions justified and supported by the results? Are hypotheses and speculations carefully identified as such?

Yes, the results are interpreted well, and the conclusions are justified by the results.

Quality of Presentation: Is the article written in an appropriate way? Are the data and analyses presented appropriately? Are the highest standards for presentation of the results used?

Yes, the article is written well, and the data are presented appropriately.

Scientific Soundness: is the study correctly designed and technically sound? Are the analyses performed with the highest technical standards? Are the data robust enough to draw the conclusions? Are the methods, tools, software, and reagents described with sufficient details to allow another researcher to reproduce the results?

Yes, study design, methods and data analysis are explained well in this manuscript.

Interest to the Readers: Are the conclusions interesting for the readership of the Journal? Will the paper attract a wide readership, or be of interest only to a limited number of people? (please see the Aims and Scope of the journal)

Yes, this would be a great article for the researchers in the breast cancer research field.

Overall Merit: Is there an overall benefit to publishing this work? Does the work provide an advance towards the current knowledge? Do the authors have addressed an important longstanding question with smart experiments?

Yes. This study provides an advancement to the current knowledge. 

English Level: Is the English language appropriate and understandable?

Yes, English language in the manuscript is appropriate and understandable. 

Overall Recommendation: Accept after Minor Revisions

This is a well written paper and I only have few comments that might be helpful to improve the manuscript. 

Abstract

Abstract is well written and summarizes the content of the manuscript.

Introduction

Introduction is well written and explains the about BRCA 1 and BRCA 2 mutations and importance of these genes in DNA repair pathway.

In the end of the introduction, authors could briefly mention the methods used in this study.

Results

Line 130: “BRCA1-deficient HCC1937 (HCC) cells that showed residual BRCA1 expression were used as negative controls.”

Briefly explain what type of cell line HCC1937 is? Other than BRCA1 deficiency is there a special reason to select this cell line as a negative control?

Why are two different units used for the concentration of MMC and Talazoparid? It makes it difficult to compare the results.

Tables, Figures and Figure legends

Figure 1: The scale on Figure 1E is not clear.

Figure 3: Insert scale bar for Figure 1B, and the labels on the Figure 1B is not clear.

Materials and Methods:

Line 542: Cells were washed in PBS + 0.1 % Tween20 and stained with propidium iodid (10 μg/ml with 1 % RNase and 0.1 % Triton X-100) for 30 min in the dark.

Correct the typo in this sentence, “iodid”

References:

Most of the references (17 out of 57) are more than 10 years old. Please replace these with new references if possible.

Author Response

Reviewer comment: This is a well written paper and I only have few comments that might be helpful to improve the manuscript.

Answer: We thank the reviewer for the positive assessment of our manuscript

Reviewer comment: In the end of the introduction, authors could briefly mention the methods used in this study.

Answer: We thank the reviewer for this helpful comment and now introduced the two sentences “To this end, BRCA1 was modified by CRISPR/Cas9-mediated genome editing and con-sequent effects on DNA repair processes were assessed by the plasmid reconstruction assays, visualized by DNA repair markers such as RAD51, yH2AX and RPA by immunofluorescence. By Western blot the expression and activation of DNA damage response proteins were measured. The effects on replication processes were analyzed by DNA fiber assay, cell cycle distribution by FACS analysis, and cellular survival after MMC, talazoparib, and irradiation by colony formation assay.”, at the end of the introduction.

Reviewer comment:  Line 130: “BRCA1-deficient HCC1937 (HCC) cells that showed residual BRCA1 expression were used as negative controls.”

Briefly explain what type of cell line HCC1937 is?

Answer: We thank the reviewer for this helpful comment and have now added the following sentence, “The lymphoblast cell line HCC1937 isolated from a BRCA1 germline mutation carrier showing residual BRCA1 expression was used for comparison [27, 28].”

Reviewer comment: Other than BRCA1 deficiency is there a special reason to select this cell line as a negative control?

Answer to reviewer: No, in fact, the well-described BRCA1 mutation of the cell line was the main reason to use it as a control. We were initially surprised to observe residual protein in our approach, despite mutations in multiple alleles, and therefore chose an established, commercially available BRCA1 deficient cell line with a proven HR defect as control. Despite mutations in both alleles and HR deficiency, this also shows residual BRCA1 protein activity. We can only explain this by the fact that a residual activity of BRCA1 must be maintained despite mutation in order to ensure survival.

Reviewer comment: Why are two different units used for the concentration of MMC and Talazoparid? It makes it difficult to compare the results.

Answer: We excuse for this mistake and have now equalized the unit for the concentration of MMC and talazoparip.

Reviewer comment: Figure 1: The scale on Figure 1E is not clear. Figure 3: Insert scale bar for Figure 1B, and the labels on the Figure 1B is not clear.

Answer: In both figures the ambiguities have been removed.

Reviewer comment: Materials and Methods: Line 542: Cells were washed in PBS + 0.1 % Tween20 and stained with propidium iodid (10 μg/ml with 1 % RNase and 0.1 % Triton X-100) for 30 min in the dark. Correct the typo in this sentence, “iodid”

Answer: The typo is now corrected to iodide

Reviewer comment: Most of the references (17 out of 57) are more than 10 years old. Please replace these with new references if possible.

Answer: We thank the reviewer for this helpful hint and have replaced or supplemented references with more recently dated ones. Anyway, we would like to keep citations referring to important original observations.

Round 2

Reviewer 1 Report

The authors have addressed and properly discussed all the raised concerns.